# A Novel DC-AC Fast Charging Technology for Lithium-Ion Power Battery at Low-Temperatures

**Shanshan Guo** [1], **Zhiqiang Han** [2], **Jun Wei** [3,*], **Shenggang Guo** [4] **and Liang Ma** [5,*]

1. School of Electromechanical and Vehicle Engineering, Weifang University, Weifang 261061, China; guoss@wfu.edu.cn
2. School of Mechanical Engineering and Automation, Northeastern University, Shenyang 110057, China; 17853681013@163.com
3. School of Mechanical, Tianjin University of Technology and Education, Tianjin 300355, China
4. School of Vehicle and Mobility, Tsinghua University, Beijing 100084, China; lucy001478@163.com
5. School of Mechanical Engineering, Ningxia University, Yinchuan 750014, China
* Correspondence: wj18651218159@163.com (J.W.); ml_bit@126.com (L.M.)

**Abstract:** There are several drawbacks for lithium-ion batteries at low temperatures, including weak electrolyte conductivity, low chemical reaction rate and greatly increased impedance. Thus, it is inefficient to charge lithium-ion batteries at low temperatures. This work proposes an AC incentive fast charging strategy at low-temperatures for lithium-ion batteries based on the analysis and comparison of the existing charging and heating methods. The charging speed, temperature variation, the capacity loss of the constant current constant voltage (CCCV) charging strategy and the proposed method with different current and frequency conditions are compared and analyzed. The results show that it takes about 1400 s for the proposed method to fully charge a lithium-ion battery in the case of 2.2 A current beginning at 25% state of charge (SOC). In addition, the temperature rises about 8 °C. In contrast, the charging time of the CCCV method is 400 s slower than the proposed method and the temperature of the CCCV method increases only about 2 °C. In the case of 1.5 A current beginning at 0% SOC, the charging time of the proposed method is 500 s faster than the CCCV method. The results indicate that the proposed charging method can significantly improve the charging efficiency of lithium-ion batteries at low temperatures.

**Keywords:** lithium-ion batteries; low-temperature charging; heating at low-temperature; fast charging

## 1. Introduction

Electric vehicles are regarded as a response to the era of energy conservation and environmental protection by solving the problem of energy loss and environmental pollution [1–3]. Lithium-ion batteries have become the main energy supplier of electric vehicles because of their high energy density, excellent durability and reliability [4–6]. However, the lithium-ion batteries have some drawbacks at low temperatures. For instance, the electrolyte ionic conductivity of lithium-ion batteries is weak [7,8], the chemical reaction rate is low for the battery electrolyte [9–12], and the conductivity of SEI film on the surface of negative graphite particles is weak [13]. In addition, the diffusion coefficient of lithium-ion in the anode graphite particles is relatively low at low temperatures [14–16], which results in slow charging. It is also easy to accumulate lithium ions on a negative surface during charging and forming lithium metal at low temperatures, which will cause permanent damage to the battery after a long-time growth of lithium metal dendrites [17–24]. Therefore, it is meaningful to investigate a new approach to address the aforementioned issues in the case of batteries operated at low temperatures.

### 1.1. Literature Review

Several methods for charging at low temperatures have been explored in previous studies, and can be divided into two main categories: external heating and internal heating. The main external heating methods are warm air heating, liquid heating, heating film heating and circulating high temperature gas heating. [25,26]. The warm air heating method is simple and cheap but inefficient in comparison with the liquid heating method [27,28]. The liquid heating method is similar to the circulating high temperature gas heating, but the liquid heating effect is better than gas heating because of the higher liquid thermal conductivity. The liquid heating method generally heats batteries through the heat exchange between the liquid and the batteries. The commonly used heat transfer mediums are oil, glycol and water. This method is implemented by immersing the batteries into the liquid or the pipeline around the battery. However, this method requires the battery or the battery box to be highly airtight, which increases the difficulty of the battery designing. Compared with the liquid heating method, arranging heating plates on the surface of the power battery has the advantages of higher heating efficiency and simple structure of the heating device [29–32]. The disadvantages are uneven heating, long heating time, excessive temperature difference between the inside and outside of the cell and high accuracy of temperature control requirements. External heating becomes less important due to low heating rate, low energy utilization efficiency and uneven temperature distribution. Researchers have begun to study internal heating methods, which mainly include internal self-healing method, internal resistance heating, convection heating, pulse heating and alternating current (AC) excitation heating. [33–41]. The self-heating method preheats the battery by charging and discharging the battery itself to raise the temperature at low temperatures. Wu et al. [42] used the constant current discharging to heat the battery and the results indicate that the internal temperature of the battery is more uniformly distributed and heats up faster than the external temperature. However, this method consumes more energy and accelerates battery aging. Ji et al. [43] proposed an MPH method to preheat the battery that consumes less energy compared to the internal heating method. However, it accelerates the battery aging and greatly complicates the system structure. Self-heating li-ion battery is a new kind of battery structure of "full temperature battery" [44], where the metal nickel sheet is used as a thermal element inside the battery. The battery is warmed up via the heat generated by the nickel sheets during the period of charging and discharge. This method has the merits of efficient heating, low energy consumption and rapid charging speed [45–48].

### 1.2. Contributions of the Work

This study proposes an AC heating excitation strategy to heat the battery in a short time during charging to eliminate the negative effect caused by the low temperature situation. The main contributions of this work are summarized as follows:

(1) An AC heating excitation strategy is designed and implemented to realize the temperature increases to a condition that is suitable for battery charging.
(2) The selection of frequency and charging rate are investigated to obtain the optimal parameters of the charging experiment.
(3) The charging efficiency and the loss of capacity between the proposed method and the constant current constant voltage (CCCV) method are compared quantitatively.

### 1.3. Structure of the Work

The rest of this paper is organized as follows. Section 2 introduces the characteristics of the AC incentive charging strategy in detail. Section 3 presents the experimental setup and experimental methods to verify the charging strategy. Section 4 discusses the results of the charging strategy and Section 5 summarizes the main conclusions.

## 2. Main Parameters of the AC Incentive Charging Strategy

*2.1. Requirement of the AC Incentive Charging Strategy at Low-Temperature Environment*

Lithium-ion batteries that charge quickly in low-temperature environments require a short charging time and a rapid temperature rise without sacrificing battery life and safety. To achieve this goal, the following two conditions should be met.

Condition I: The negative region should have a large solid-liquid potential difference during the charging process of lithium-ion batteries, where the total polarization voltage of the battery should be small to inhibit lithium plating [49]. This process can be described as follows:

$$E_{\text{eq,neg}}(c_{s,r_p}/c_{s,max}) - |\eta_{\text{neg}}| - |\triangle\phi_{\text{film,neg}}| - \phi_{\text{l}} > 0 \tag{1}$$

where $E_{\text{eq,neg}}$ is the equilibrium potential of the negative electrode, $c_{s,\text{rp}}$ is the concentration of lithium ions on the particle surface of the negative electrode, $c_{s,\text{max}}$ is the maximum concentration of lithium ions in the negative particles, $\eta_{\text{neg}}$ is the electrochemical over-potential of the negative electrode, $\triangle\phi_{\text{film,neg}}$ is film impedance voltage drop of the negative electrode and $\phi_{\text{l}}$ is liquid phase potential.

Condition II: The upper and lower cutoff voltage is higher than the voltage at both ends of the lithium-ion battery [50]. This condition can be described as follows:

$$U_{\text{ct}} = I\left(\frac{(U_{\text{t,max}} - U_{\text{OCV}})}{\left| R_{\text{i}} + \frac{R_{\text{SEI}}}{(j2\pi f)^{n_{\text{SEI}}} Q_{\text{SEI}} R_{\text{SEI}} + 1} + \frac{R_{\text{ct}}}{(j2\pi f)^{n_{\text{ct}}} Q_{\text{ct}} R_{\text{ct}} + 1} \right|}\right) < |U_{\text{t,terminal}} - U_{\text{OCV}}| \tag{2}$$

where $U_{\text{t,max}}$ is the upper cut-off voltage, $U_{\text{ct}}$ is the polarization voltage, $I$ is current, $U_{\text{OCV}}$ is the open-circuit voltage, $U_{\text{t,terminal}}$ is the cut-off voltage, $R_{\text{i}}$ is the $O_{\text{hm}}$ resistance, $R_{\text{SEI}}$ is the resistance of the SEI film, $R_{\text{ct}}$ is the polarization resistance, $Q_{\text{SEI}}$ and $Q_{\text{ct}}$ are the two constant phase elements (CPEs), $n_{\text{SEI}}$ and $n_{\text{ct}}$ is the exponential factor and $f$ is the excitation frequency.

*2.2. Selection of Frequency and Current*

The mathematical equation for the battery-electric heating process is:

$$q_{\text{Total}} = q_{\text{AC}} + q_{\text{DC}} = \left(\frac{U_{\text{max}} - U_1}{\sqrt{2}}\right)\frac{R_{\text{re}}}{|Z|^2} + \frac{(U_{\text{oc}} - U_1)^2}{|Z|} \tag{3}$$

where $q_{\text{Total}}$ is the total thermogenic rate of the power battery, $q_{\text{AC}}$ is the thermogenic rate of AC, $q_{\text{DC}}$ is the thermogenic rate of DC, $R_{\text{re}}$ is the real impedance part of the power battery and $Z$ is the total impedance of the power battery.

The excitation frequency can be calculated as follows:

$$\left\{ \begin{array}{l} a = \dfrac{\dfrac{\frac{1}{R_{\text{SEI}}} + Q_{\text{SEI}}(2\pi f)^{n_{\text{SEI}}} \cos\frac{\pi n_{\text{SEI}}}{2}}{\left(\frac{1}{R_{\text{SEI}}}\right)^2 + \frac{2}{R} Q_{\text{SEI}}(2\pi f)^{n_{\text{SEI}}} \cos\frac{\pi n_{\text{SEI}}}{2} + Q_{\text{SEI}}{}^2 (2\pi f)^{2n_{\text{SEI}}}} + R_{\text{i}}}{\left| R_{\text{i}} + \frac{R_{\text{SEI}}}{(j2\pi f)^{n_{\text{SEI}}} Q_{\text{SEI}} R_{\text{SEI}} + 1} + \frac{R_{\text{ct}}}{(j2\pi f)^{n_{\text{ct}}} Q_{\text{ct}} R_{\text{ct}} + 1} \right|^2} \\[30pt] b = \dfrac{\dfrac{\frac{1}{R_{\text{ct}}} + Q_{\text{ct}}(2\pi f)^{n_{\text{ct}}} \cos\frac{\pi n_{\text{ct}}}{2}}{\left(\frac{1}{R_{\text{ct}}}\right)^2 + \frac{2}{R} Q_{\text{ct}}(2\pi f)^{n_{\text{ct}}} \cos\frac{\pi n_{\text{ct}}}{2} + Q_{\text{ct}}{}^2 (2\pi f)^{2n_{\text{ct}}}}}{\left| R_{\text{i}} + \frac{R_{\text{SEI}}}{(j2\pi f)^{n_{\text{SEI}}} Q_{\text{SEI}} R_{\text{SEI}} + 1} + \frac{R_{\text{ct}}}{(j2\pi f)^{n_{\text{ct}}} Q_{\text{ct}} R_{\text{ct}} + 1} \right|^2} \\[20pt] F(f) = a + b \end{array} \right\} \tag{4}$$

where $R_{\text{i}}$ is the ohmic internal resistance, $R_{\text{ct}}$ is the polarization impedance, $R_{\text{SEI}}$ is the SEI impedance, $Q_{\text{ct}}$ is the polarization partial constant phase element, $Q_{\text{SEI}}$ is the SEI partial constant phase element, $n_{\text{ct}}$ is the polarization partial constant phase component index, $n_{\text{SEI}}$ is the SEI partial constant phase element index and $f$ is the frequency.

The optimal excitation current is calculated as follows:

$$i_{\text{opt}} = \frac{(U_{\text{t,max}} - U_1)/\sqrt{2}}{\left| R_{\text{i}} + \frac{R_{\text{SEI}}}{(j2\pi f_{\text{opt}})^{n_{\text{SEI}}} Q_{\text{SEI}} R_{\text{SEI}} + 1} + \frac{R_{\text{ct}}}{(j2\pi f_{\text{opt}})^{n_{\text{ct}}} Q_{\text{ct}} R_{\text{ct}} + 1} \right|} + \frac{(U_{\text{oc}} - U_1)}{\left| R_{\text{i}} + \frac{R_{\text{SEI}}}{(j2\pi f_{\text{opt}})^{n_{\text{SEI}}} Q_{\text{SEI}} R_{\text{SEI}} + 1} + \frac{R_{\text{ct}}}{(j2\pi f_{\text{opt}})^{n_{\text{ct}}} Q_{\text{ct}} R_{\text{ct}} + 1} \right|} \tag{5}$$

The excitation current is the current flowing through the ohmic internal resistance, $U_{\text{t,min}}$ is the end voltage, $U_{\text{t,max}}$ is the upper limit of the allowable end voltage, $U_{\text{oc}}$ is the open-circuit voltage and $f_{\text{otp}}$ is the optimal excitation frequency.

## 3. Experiments

### 3.1. Experiment Setup

The charging speed and the temperature rise speed of the LG INR18650HG2 battery were used to evaluate the performance of the AC excitation charging strategy. The characteristics of the test battery are listed in Table 1.

**Table 1.** The parameters of the test battery.

| Brand Model | LG INR18650HG2 |
| --- | --- |
| Standard capacity | 3000 mAh |
| Rated voltage | 3.6 V |
| Charging voltage | 4.2 ± 0.05 V |
| Battery charging | Standard 1500 mA; cutoff current 50 mA(cc-cv) |
| Fast charge | 4000 mA, cutoff current 100 mA(cc-cv) |
| Maximum continuous discharge current | 20 A |
| Discharge cut-off voltage | 2.5 V |
| Essential resistance (AC 1 KHz) | ≤20 mΩ, with our PTC |

The experimental devices are displayed in Figures 1 and 2. The external temperature environmental conditions were provided by a thermal chamber. The BTS 600 was used to calibrate the capacity of the battery by the constant current and constant voltage charge and discharge. The AC power was provided by an inverter. A computer was used to control the battery charge and discharge process and record the data using a procedure.

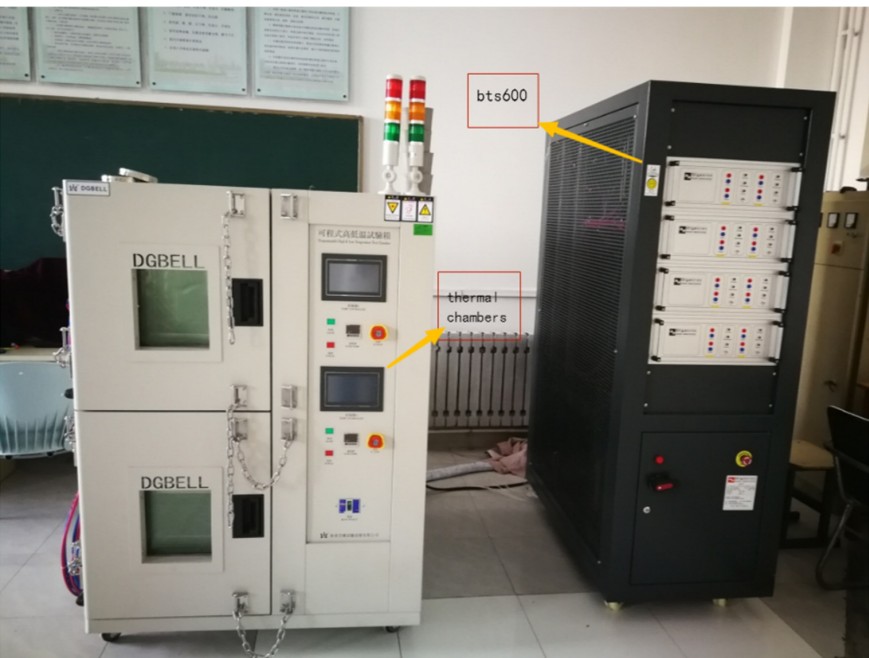

**Figure 1.** The BTS600 and thermal chamber.

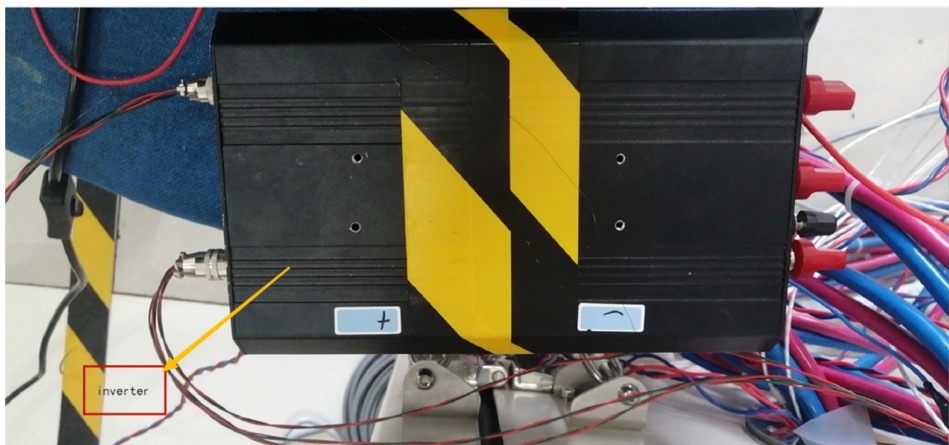

**Figure 2.** The AC inverter.

The test content of this experiment mainly included temperature setting, capacity test, SOC calibration and charging test at low temperatures. Several variables were observed and recorded during the AC and CCCV charging and discharging process, including the rise rate of the temperature, the speed of charging, the change of voltage and the capacity loss. The cut-off voltages during the charging and discharging period were 4.2 V and 2.5 V, respectively. A temperature sensor was fitted on the middle of the battery surface to monitor the change of the temperature. The voltage and current were monitored by sensors and recorded by the computer in real time.

### 3.2. Test Scheme

The battery was charged via AC incentive strategy by the inverter at the specified temperature and SOC. The external ambient temperature was set to 0 °C and 5 °C to evaluate the effect caused by the temperature It should be noted that the battery should be kept stationary for more than two hours in each environment to ensure the battery temperature reaches the external ambient temperature in the laboratory.

The experiment was implemented at different temperatures and different SOC, including CCCV charge and discharge tests and AC charging tests. The CCCV charging method was used to charge the battery to the specified SOC point. Then the temperature was adjusted to the specified value by the thermal chamber. After a stand of two hours, the battery inside the thermal chamber reached the required temperature. Subsequently, the AC and CCCV strategies were used to charge the battery until the voltage reached 4.2 V. The temperature voltage and current were recorded during the experiment. In this study, the initial SOC of the AC and CCCV charging strategies were 0%, 25% and 50%. The charging rates were set to 0.5 C, 0.6 C and 0.8 C. The tests were implemented at 0 °C and 5 °C.

## 4. Results and Discussion

In order to study the AC excitation charging strategy at different temperatures and general applicability, the charge and discharge experiments were carried out under conditions of 0 °C and 5 °C temperature, 20% SOC and 50% SOC respectively.

### 4.1. The Influence of Temperature on the Charging Speed

Taking the 0% SOC battery as an example, as shown in Figure 3, there were different temperature rise rates at different external temperatures. Firstly, the battery temperature rise rate at 5 °C environment was faster than the 0 °C. Secondly, as the temperature rose, the temperature rise rate tended to be stable. Finally, the final battery temperature was higher than the 0 °C battery temperature at the 5 °C environment condition.

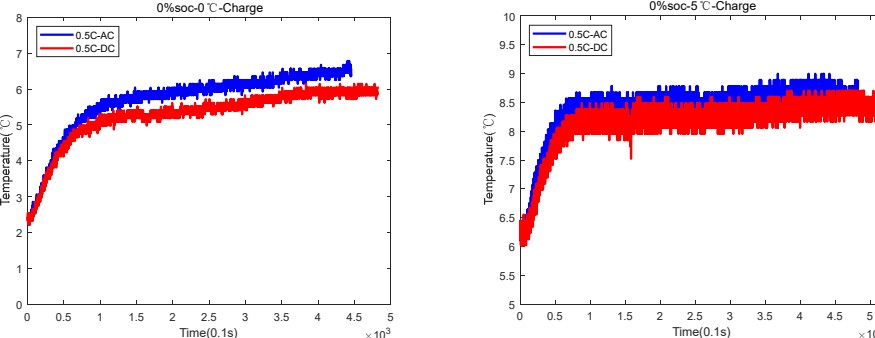

**Figure 3.** The temperature in the case of charging starting at 0% SOC at 0 °C and 5 °C respectively.

It can be seen that the lower the temperature, the lower the temperature rise rate; by contrast, the lower the temperature, the larger the temperature change range.

It can be seen from Figure 4 that the battery voltage rose greatly at the beginning of charging. The rate of rise gradually accelerated as charging proceeded. In the case of charging start at 0% SOC, the 5 °C DC charging time was 200 s faster than 0 °C DC charging time while the 5 °C AC charging time was 100 s faster than 0 °C DC charging time. This indicates that the AC charging strategy performed more efficiently than the DC charging strategy.

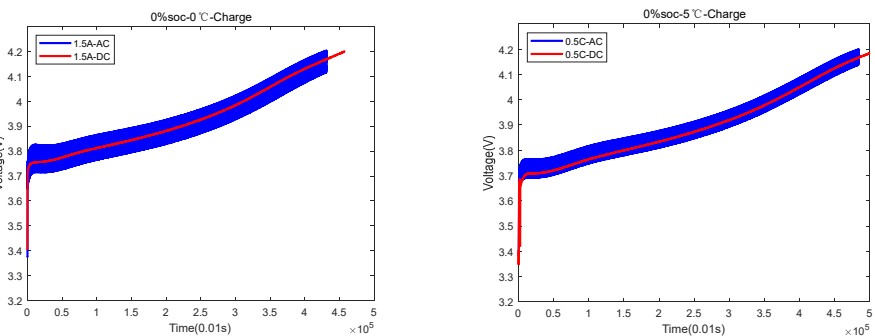

**Figure 4.** The voltage changes in the case of charging starting at 0% SOC at 0 °C and 5 °C, respectively.

### 4.2. Effects of Initial SOC on Charging Speed

This section introduces battery charging with different initial SOC and different temperatures.

As shown in Figure 5, the initial capacity of the battery was 25% and it was charged at an external environment of 5 °C. The temperature change graph is shown on the left side, indicating that the battery temperature rises rapidly during the initial stage of charging. The slope of the curve for AC charging was greater than that for DC charging. The curve tended to be flat after 1000 s, probably because the heat generated by the battery charging was the same as that emitted by the battery. It changed back and forth between 9 °C and 10 °C overall. The voltage change graph is on the right side. It is obvious that the voltage rose instantly at the beginning of charging. The voltage curve of AC charging was steeper than that of DC charging on the whole. In the early stage of charging, the charging speed was slow, the voltage rose more slowly, and the curve was flat. In contrast, the charging speed was accelerated and the slope of the curve became larger in the later charging stages, which may be caused by the rise of temperature and the decrease of the internal resistance of the battery. The AC charging took 2700 s and the DC charging took 3800 s, an increase of 28%.

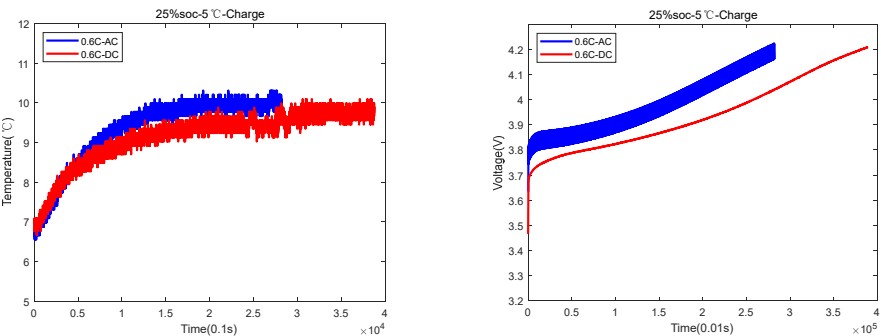

**Figure 5.** The temperature and voltage changes in the case of charging starting at 25% SOC and 5 °C.

For Figures 6–8, the temperature changes and voltage changes are similar to those in Figure 5.

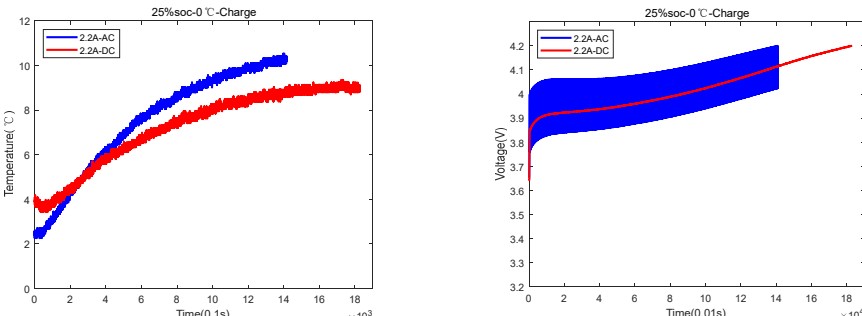

**Figure 6.** 0 °C and 25% SOC temperature and voltage change diagram.

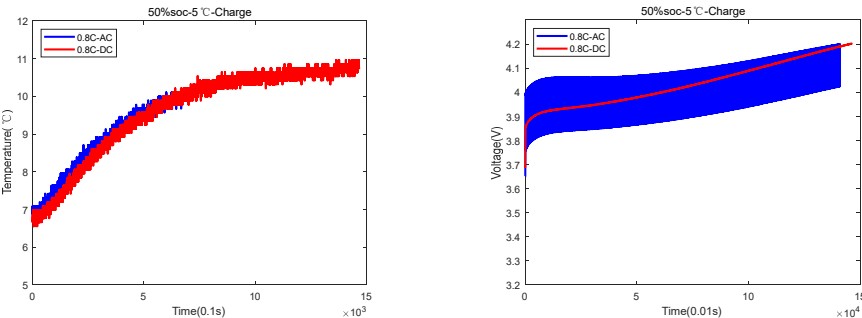

**Figure 7.** 5 °C and 50% SOC Temperature and voltage change diagram.

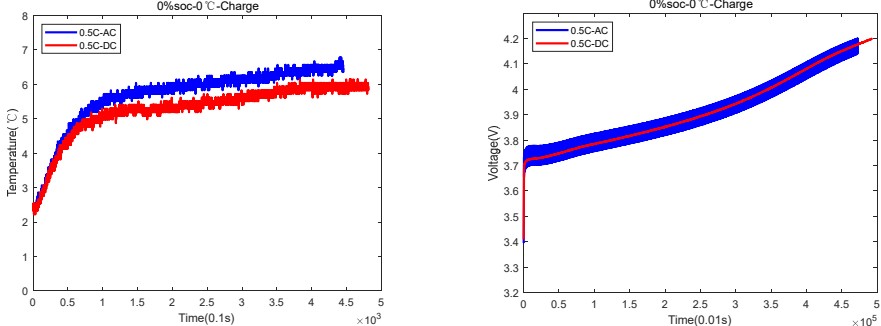

**Figure 8.** 0 °C and 0% SOC temperature and voltage change diagram.

As shown in Figure 6, when the initial battery capacity was 25%, the external temperature was set to 0 °C, and the charging current was set to 2.2 A, it can be seen that the temperature rose rapidly at the beginning of AC charging, after which the temperature rise speed slowed down subsequently. The overall charging time of the AC strategy was 1300 s while that of the DC strategy was 1800 s, where the AC charging speed rose by 22%

compared with the DC charging. The temperature of AC charging rose by 8 °C while that of the DC strategy rose by 5 °C.

As shown in Figure 7, when the initial battery capacity was 50%, the external temperature was 5 °C, and the charging current was 0.8 c, the AC full charging time was 1300 s while the DC full charging time was 1500 s. The charging speed was increased by 13%.

As shown in Figure 8, when the initial battery capacity was 0%, the battery external temperature was set to 0 °C, and the current user was set to 0.5 c, the AC full charging time was 4300 s while the DC full charging time was 4900 s. The charging speed was increased by 12%.

The specific temperature changes and the time required for charging are shown in Table 2.

**Table 2.** Temperature and time.

|  | **Charging Time** | **Temperature Variation** |
|---|---|---|
| 0%SOC-0 °C | 4300 s AC and 4800 s DC | AC temperature rose by 5 °C, and DC temperature rose by 3.5 °C |
| 0%SOC-5 °C | 4700 s for AC and 5200 s for DC | AC temperature rose by 3 °C, and DC temperature rose by 2.5 °C |
| 25%SOC-0 °C | 1300 s for AC and 1800 s for DC | AC temperature rose by 8.5 °C, and DC temperature rose by 5 °C |
| 25%SOC-5 °C | 2700 s for AC and 3800 s for DC | AC temperature rose by 3.5 °C, and DC temperature rose by 3 °C |
| 50%SOC-5 °C | 1400 s for AC and 1500 s for DC | AC temperature rose by 6 °C, and DC temperature rose by 4.5 °C |

## 5. Conclusions

In this paper, a low-temperature fast charging strategy was investigated using the AC excitation charging method. The system performed DC and AC charging and discharging experiments with different initial SOCs at different temperatures to monitor the temperature change, voltage change and charge power. The main conclusions can be summarized as follows: (1) The AC excitation charging strategy is faster than that of CCCV method by 20%. (2) The AC excitation charging strategy has a fluctuating voltage rise compared to the CCCV method. As a result, the temperature rises faster, with the average temperature increasing 2 to 4 °C compared to the CCCV method. Therefore, the proposed charging scheme can significantly reduce the charging time at a reasonable temperature rise, which is superior to the conventional CCCV method.

In future practical applications, the AC excitation charging strategy can be used for fast charging of electric vehicles in low temperature environments.

**Author Contributions:** Conceptualization, S.G. (Shenggang Guo); Data curation, J.W.; Formal analysis, J.W.; Funding acquisition, S.G. (Shanshan Guo); Investigation, Z.H.; Methodology, S.G. (Shanshan Guo); Project administration, S.G. (Shenggang Guo); Resources, S.G. (Shenggang Guo); Software, J.W.; Supervision, S.G. (Shanshan Guo); Validation, L.M.; Writing—original draft, J.W.; Writing—review & editing, Z.H. and L.M. All authors have read and agreed to the published version of the manuscript.

**Funding:** This work was supported by the National Natural Science Foundation of China (Grant 5210070756). The systematic experiments on the lithium-ion batteries were performed at the New Energy Vehicle Power System Engineering laboratory, Weifang University.

**Institutional Review Board Statement:** Not applicable.

**Informed Consent Statement:** Not applicable.

**Conflicts of Interest:** The authors declare no conflict of interest. The funders had no role in the design of the study; in the collection, analyses, or interpretation of data; in the writing of the manuscript, or in the decision to publish the results.

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
