# Peer review of "A Novel DC-AC Fast Charging Technology for Lithium-Ion Power Battery at Low-Temperatures"

_sustainability, doi:10.3390/su14116544_

Round 1

Reviewer 1 Report

This article presents a novel DC-AC fast charging technology for lithium-ion power battery at low-temperatures. Article is well written, the results are well presented and discussed. Some comments and suggestions to better improve the article are given here:
Please check equation 4.
What is the limit of your used technology?
The conclusion must be improved.
Organize the references on the same style, for example use the APA style.

Reviewer 2 Report

After reading the submitted paper by Shanshan et al., I have the following suggestions:
1- The first sentence in the Abstract need to be rewritten in better way
2-  The abstract need more strengthen by supporting with some important results.
3- Add more keywords.
4- Some references need to be cited in the introduction:
https://doi.org/10.1016/j.cej.2022.136469
https://doi.org/10.1016/j.cej.2021.131185
DOI: 10.1039/d0se01493a

5- Authors should add a comparison with some related works 

Reviewer 3 Report

In this paper, the AC incentive charging method is adopted to study the low-temperature fast charging strategy. The system carries out DC and AC charging and discharge experiments on different initial soc batteries at different temperatures to monitor the temperature change, voltage change, and charging amount. It is shown that AC incentive charging strategy is faster, and the average charging time is reduced by 20% than the CCCV method. (2) AC incentive charging causes voltage fluctuations to rise, compared with voltage rise steadily using the CCCV method. Thus, the temperature rises faster, and which average temperature is increased by 2~4℃ than the CCCV method.

This paper contains new enough results. Suggestions:

1) The English should be improved

2) More information is needed in the figure caption "Fig.6 0℃ and 25%soc temperature and voltage change diagram"

3) The effect of charging method on charging speed is not clear

4) In page 4 line 79,  the authors say that " The 79 metal nickel sheet is used as a thermal element in the battery inside, during the period of charging and discharge, the current warm up the battery through the nickel sheets to generate heat. The method has significant improvements, including efficient heating, low energy consumption, and rapid charging speed [.....].  The authors may cite the following papers:

Vernardou, D., Kazas, A., Apostolopoulou, M.,  Katsarakis, N., E. Koudoumas, E., Hydrothermal Growth of MnO2 at 95 oC as an Anode Material, Int. J. of Thin Film Science and Technology  5 (2016), pp. 121-127

Al-Qrinawi, M. S.,  El-Agez, T. M.,  Abdel-Latif, M. S.,  Taya, S. A., Capacitance-voltage measurements of hetero-layer OLEDs treated by an electric field and thermal annealing, Int. J. of Thin Film Science and Technology  10 (2021), pp. 217-226

Elhadary, A. A.,  El-Zein, A.,  Talaat, M., El-Aragi, G.,  El-Amawy, A., Studying The Effect of The Dielectric Barrier Discharge Non- thermal Plasma on Colon Cancer Cell line, Int. J. of Thin Film Science and Technology  10 (2021), pp. 161-168

Varun Joshi, Mamta Kapoor, A Novel Technique for Numerical Approximation of 2 Dimensional Non-Linear Coupled Burgers’ Equations using Uniform Algebraic Hyperbolic (UAH) Tension B-Spline based Differential Quadrature Method, Appl. Math. Inf. Sci. 15, (2021) PP: 217-239: doi:10.18576/amis/150215

Reviewer 4 Report

1.  Extensive editing of English language and style required.

2. Some of the sentences in the introduction are too long and need to be paraphrased like the first sentence in the lines 30-35.

3. The sources of the two conditions mentioned in Section 2.1 on page 6 or any theoretical evidence for them are not mentioned. The authors should provide more explanation and documentation for these two conditions.

4. Descriptive equation of the optimal excitation current (Equation 5) in page 8 needs more clarification.

5.  The data needs to be described more comprehensively in Experiments section, especially Tables should be made and explained based on the information studied and their effects.

6. Future Scope is missing please include it in concluding remarks.

Reviewer 5 Report

This manuscript investigated the effects of a DC_AC method on the fast charging performance of Li-ion batteries. The authors set up a test rig which is able to customise the operating temperature and charging rate. A series of studies with respect to the operating parameters of Li-ion batteries were further conducted. Below list my comments:

  1. The English really needs substantial improvements. Please carefully look through the manuscript and correct all typos, e.g. Line 36, Line 42, Line 43 and so on.
  2. In the section “contribution of this work”. I strongly suggest authors rewrite it, as the current version is obviously not coherent. The most important point is to highlight your novelty and the significant difference between your work and others.
  3. In the Section of results and discussion, the English are terrible that don’t coherently deliver the main thoughts of the figures. Moreover, deeper analysis and discussion are strongly suggested to extract more meaningful points.

Round 2

Reviewer 4 Report

The authors have completely replies to my questions, and they have followed my recommendations. The paper has been improved so I recommend it for publication, as it is.

Reviewer 5 Report

My comments have been addressed by the authors. I suggest it published as it is.